# Dietary Oxalate Intake and Kidney Outcomes

**DOI:** 10.3390/nu12092673

**Published:** 2020-09-02

**Authors:** Matteo Bargagli, Maria Clarissa Tio, Sushrut S. Waikar, Pietro Manuel Ferraro

**Affiliations:** 1U.O.C. Nefrologia, Dipartimento di Scienze Mediche e Chirurgiche, Fondazione Policlinico Universitario A. Gemelli IRCCS, 00168 Roma, Italy; matteo.bargagli@unicatt.it; 2Università Cattolica del Sacro Cuore, 00168 Roma, Italy; 3Division of Renal Medicine, Department of Medicine, Brigham and Women’s Hospital, Boston, MA 02101, USA; mctio@bwh.harvard.edu; 4Renal Section, Department of Medicine, Boston University School of Medicine and Boston Medical Center, Boston, MA 02101, USA; swaikar@bu.edu

**Keywords:** acute kidney injury, chronic kidney disease, diet, nephrocalcinosis, nephrolithiasis, oxalate

## Abstract

Oxalate is both a plant-derived molecule and a terminal toxic metabolite with no known physiological function in humans. It is predominantly eliminated by the kidneys through glomerular filtration and tubular secretion. Regardless of the cause, the increased load of dietary oxalate presented to the kidneys has been linked to different kidney-related conditions and injuries, including calcium oxalate nephrolithiasis, acute and chronic kidney disease. In this paper, we review the current literature on the association between dietary oxalate intake and kidney outcomes.

## 1. Introduction

Oxalate is both a plant-derived molecule and a terminal toxic metabolite with no known physiological function in humans. It is predominantly eliminated by the kidneys through glomerular filtration and tubular secretion. Intestinal secretion of oxalate also contributes to its elimination, but to a much lesser extent [1,2,3,4,5]. Urinary oxalate excretion is in part determined by endogenous synthesis through hepatic metabolism of amino acids [6] (including glycine, phenylalanine, and tryptophan), hydroxyproline [7], and net intestinal absorption from dietary sources [1].

The normal intestinal absorption of oxalate is around 10–15% [8], but varies depending on diet. Low calcium diets, for instance, have been linked to an increased absorption of oxalate [9]. Similarly, in individuals with small bowel malabsorption syndrome, oxalate uptake is increased because of a reduction in free intestinal ionized calcium [10]. In addition, gut microbiome dysbiosis might affect intestinal oxalate availability and thus its urinary excretion [11].

Regardless of the cause, the increased load of dietary oxalate presented to the kidneys has been linked to different kidney-related conditions and injuries. The most studied consequence of increased urinary oxalate excretion is calcium oxalate nephrolithiasis, a condition caused by elevated urinary supersaturation for calcium oxalate, followed by crystal formation and deposition [1,2]. Hyperoxaluria has been also associated with both acute and chronic kidney disease [12,13]. The underlying mechanisms for kidney disease include crystal tubular deposition and tubular epithelium damage [14]. In addition, the accumulation of calcium-oxalate crystals in renal parenchyma seems to generate an inflammatory response, worsening kidney injury [15]. Accumulating evidence from animal and human studies suggests that dietary oxalate may have a greater impact on kidney function than previously recognized. In this review, we discuss dietary oxalate as an important determinant of urinary oxalate excretion and its subsequent association with kidney stones, acute kidney injury (AKI), and chronic kidney disease (CKD).

## 2. Materials and Methods

In this review, we analyzed the available literature regarding the association between oxalate and kidney outcomes (nephrolithiasis, nephrocalcinosis, AKI, and CKD). We selected articles from the following databases: PubMed, Google Scholar, the Cochrane library, and Web of Science. The articles of interest were identified using the following search terms: (“kidney stones” OR “nephrolithiasis” OR “urolithiasis” OR “kidney calculi” OR “nephrocalcinosis” OR “CKD” OR “kidney disease” OR “renal failure” OR “kidney injury”) AND (“oxalate” OR “hyperoxaluria”). We only included articles written in English and with available full text. No restriction was made based on the type of article or publication period.

## 3. Results

### 3.1. Intestinal Handling of Oxalate

The gastrointestinal tract (GI) is the site of dietary oxalate absorption and oxalate secretion [3,16]. Therefore, net oxalate GI handling is contingent on the relative contributions of segment-specific oxalate absorption and secretion throughout the GI tract [17,18].

While the stomach has been shown to be a possible site of oxalate absorption [19,20], much of the oxalate handling in the gut occurs in the small and large intestines. In the intestine, oxalate handling occurs through the passive paracellular and active transcellular pathways [21]. Transcellular oxalate handling mainly occurs through transporter proteins encoded by the SLC26 gene family. There are six different SLC26 transporter proteins found along the GI tract, with SLC26A3 and SLC26A6 being the most defined in their roles in oxalate handling [22]. The apical transporter SLC26A6 is more abundant in the small intestine compared to the colon [23,24] and secretes oxalate into the lumen. In animal studies, the ileal segment of *SLC26A6* wild type mice demonstrated net luminal oxalate secretion while ileum from *SLC26A6* knock-out mice demonstrated oxalate absorption [25]. Moreover, *SLC26A6* knockout mice were found to develop hyperoxalemia, hyperoxaluria, and calcium oxalate stone formation [26]. Furthermore, an increased intestinal oxalate excretion mediated by higher SLC26A6 activity was recently found in a mouse model of CKD [27]. On the other hand, SLC26A3 facilitates oxalate absorption and is expressed more abundantly in the colon than in the small intestine. In *SLC26A3* knock-out mice, net oxalate secretion was observed in all segments of the GI tract with consequent reduction in urinary oxalate excretion [28].

The regulation of these transporters and the resultant net oxalate handling in the gut are contingent on various local and systemic pathways involving hormones, cytokines, and neurotransmitters. An in-depth review of these regulatory pathways has been discussed by Whittamore and Hatch [18].

### 3.2. Endogenous Production of Oxalate

In humans, oxalate is a terminal metabolite that is the end product of endogenous metabolism by the liver. Its immediate precursor is glyoxylate, which is either a product of hydroxyproline catabolism or oxidation of glycolate [7,29,30]. Glyoxylate is in turn trans-aminated by alanine glyoxylate aminotransferase (AGT) to form pyruvate and glycine or converted back to glycolate by glycolate/hydroxy-pyruvate reductase (GRHPR). In addition, ingestion of a variety of substances, including ethylene glycol, vitamin C or orlistat are metabolized to oxalate [31,32,33]. In primary hyperoxalurias, deficiencies of various enzymes (discussed below) lead to excess accumulation of glyoxylate which is then converted by glycolate oxidase or lactate dehydrogenase to oxalate [30].

### 3.3. Renal Handling of Oxalate

Oxalate is primarily eliminated through the kidneys by both glomerular filtration and tubular secretion [34]. Much, if not all, of kidney oxalate handling occurs in the proximal tubule when SLC26 transporters are expressed. In the mammalian kidney, SLC26A6 is the main apical chloride/oxalate exchange transporter while SLC26A1 is the basolateral sulfate/oxalate transporter [30].

Kidney handling of dietary oxalate has been extensively studied in participants with normal kidney function. While initially thought to contribute only 10–20% to urinary oxalate excretion [35], dietary oxalate is now estimated to contribute as much as 50% [1]. In a feeding study of stone formers and healthy subjects all with normal renal function, plasma oxalate was found to be controlled at a tight range after a low oxalate meal. However, the calcium stone formers exhibited higher urinary oxalate excretion with evidence of tubular oxalate secretion, signifying perhaps that tight regulation of plasma oxalate can be at the cost of increasing urinary oxalate [36]. When the dietary oxalate content is variable, the responses in urinary and plasma oxalate levels were found to be dose and time dependent post-meal. In a feeding study by Holmes et al., healthy participants were fed with increasing increments of sodium oxalate salt. Among subjects fed with 8 mmol of sodium oxalate, which is equivalent to 100 g of spinach, plasma oxalate peaked to a maximum of 3× baseline levels with urinary oxalate excretion concurrently increasing transiently to >100 mg/g Cr, levels akin to those with primary hyperoxalurias [37].

Excursions in urinary oxalate excretion are important given its contribution to kidney stone formation, and possibly, CKD progression. Oxalate feeding studies thus far have only been conducted in participants with normal kidney function. There is a current gap in the literature as to how kidney oxalate handling changes with varying degrees of kidney dysfunction.

### 3.4. Causes, Mechanisms, and Potential Treatments of Hyperoxaluria

Hyperoxaluria is defined as urinary oxalate excretion exceeding 40–45 mg/day [17], a hallmark of the primary and secondary hyperoxalurias. It should be noted, however, that substantial variability in oxalate excretion has been observed within individuals. Although higher urinary oxalate excretion increases the risk of kidney stones by increasing risk for urinary supersaturation of calcium oxalate [8], there is no definite threshold value for 24 h oxalate excretion because within-person variability is substantial and laboratory measurements are not calibrated against a common reference standard [38].

Primary hyperoxaluria (PH) is a group of devastating genetic conditions characterized by urinary oxalate excretions often higher than 100 mg/day with systemic calcium oxalate deposition resulting in cardiomyopathy, vascular disease, fractures, bone marrow suppression, nephrocalcinosis, and renal failure [39,40,41,42,43]. There are three variants, each caused by a gene mutation in an enzyme involved in the metabolism of glyoxylate, a precursor of oxalate. In PH type 1, the defective gene is AGXT, a gene that encodes for AGT [44]. This is the most common and severe variant, accounting for 80% of total cases [44]. PH types 2 and 3 are rarer, occurring in about 10% and 5% of cases respectively, whereas 5% of total PH patients have unknown mutation [45,46]. PH type 2 is caused by a dysfunction of the GRHPR enzyme due to mutations in its gene [45]. PH type 3 has recently been described and was found to be linked to a mutation in the HOGA1 gene, resulting in a defective mitochondrial 4-hydroxy 2-oxoglutarate aldolase [46]. All types of primary hyperoxalurias have poor outcomes with higher risks for CKD and end stage renal disease (ESRD) compared to the normal population [43,47,48,49]. In a registry of PH patients, ESRD was identified at a median age of 24 years [50]. In children, five-year mortality rate after renal replacement therapy initiation was 24% [47].

Secondary hyperoxaluria due to dietary oxalate alone may occur in cases of extremely elevated oxalate consumption (>1000 mg/day) [51,52]. More frequently, increased urinary oxalate excretion is found in patients with gastrointestinal surgeries, inflammatory bowel diseases, and malabsorption syndromes, collectively termed enteric hyperoxalurias [53,54]. These conditions share fat malabsorption as the underlying cause of hyperoxaluria: free fatty acids and bile salts that accumulate in the gut lumen bind to cations like calcium and magnesium, in turn increasing the solubility and intestinal absorption of oxalate [10]. In addition, both high intestinal content of bile salts and bowel inflammation are thought to cause an increase in colonic mucosal permeability leading to an increase in paracellular absorption of oxalate. Moreover, vitamin B6 deficiency may be attributable to malabsorption syndrome [55,56]. Although, in PH patients, reduced availability of pyridoxine, a cofactor of the enzyme alanine glyoxylate aminotransferase, induces glyoxylate accumulation and higher endogenous oxalate production [57], the effect of vitamin B6 supplementation in enteric hyperoxaluria is less clear.

Dietary calcium has been shown to affect urinary oxalate excretion. Early in vitro experiments have demonstrated that calcium binds to oxalate causing it to be insoluble. In vivo, increased calcium supplementation significantly reduced urinary oxalate excretion, leading to the postulate of calcium precipitating oxalate and thus preventing its absorption in the gut [58,59]. Several epidemiologic studies have also shown the inverse relationship between dietary calcium intake and urinary oxalate excretion in both healthy individuals and stone formers [58,60,61,62].

Gut microbiome may also play a role in the genesis of hyperoxaluria. In fact, *Oxalobacter formigenes*, an oxalate-degrading Gram negative bacteria, was found to be lacking in patients affected by inflammatory bowel diseases [63] and its intestinal colonization was linked to reduced urinary oxalate excretion [64]. More recently, Miller et al. underlined the effort of a network of a large number of bacterial species rather than *O. formigenes* alone in intestinal oxalate homeostasis [65]. The modification in gut microbiota diversity may be due to inflammation, antimicrobial therapies, or changes in diet [66].

Other potential etiologies of enteric hyperoxaluria are genetic variations of genes involved in intestinal oxalate transport [67]. *SLC26A6* null mice developed calcium oxalate nephrolithiasis associated with hyperoxaluria [26] and reduced SLC26A6 mRNA and protein expression was found in obesity-related hyperoxaluric mice [68].

While orthotopic liver transplantation is the only definitive cure for PH [50], numerous potential therapies for primary and enteric hyperoxalurias are currently being studied [69]. One approach to PH therapy involves downregulation of key enzymes, glycolate oxidase and lactate dehydrogenase, in the oxalate metabolism pathway through gene silencing methods [70,71,72,73] or enzyme inhibition [74]. Stiripentol [75] is a non-competitive inhibitor of lactate dehydrogenase 5 enzyme that has been shown recently to reduce hepatic production of oxalate and subsequent urinary oxalate excretion in rats and in humans [76]. A phase 2 clinical trial is currently underway, evaluating the efficacy of Stiripentol in patients with PH (https://clinicaltrials.gov/ct2/show/NCT03819647). Another approach involves oral administration of *O. formigenes* to increase gut degradation of oxalate. thereby decreasing subsequent oxalate absorption [77] (https://clinicaltrials.gov/ct2/show/NCT03116685). For enteric hyperoxaluria, oral administration of oxalate decarboxylase is currently being studied by two pharmaceutical companies with promising results [78] (https://clinicaltrials.gov/ct2/show/NCT03847090). An in-depth review of novel investigational therapies for the hyperoxalurias was recently published by Kletzmayr et al. [69].

### 3.5. Dietary Oxalate and Kidney Outcomes

Dietary oxalate intake is one of the modifiable risk factors of hyperoxaluria [79]. Although not without limitations, 24 h urine collection is the most common method for estimating daily oxalate consumption and absorption [1]. Alternatively, food frequency questionnaires have been used, although their utility in substitution to daily urine collection is debated [80]. Under physiological conditions, the main dietary factors influencing both intestinal oxalate uptake and increased endogenous production, thus increasing its urinary excretion, are foods with high oxalate content, low dietary consumption of calcium, and higher vitamin C intake [40,81]. Oxalate is present in a large variety of different foods and the exact estimation of their impact on urinary excretion is rather difficult [8]. In detail, the exact oxalate content of foods is equally determinable with both capillary electrophoresis and ion chromatography [51]. Examples of high oxalate foods are green-leaf vegetables, tea, nuts, chocolate and rhubarb [82]. The list of higher oxalate foods reported in Table 1 was obtained using already published evidence [83], including data shown by the Harvard T.H. Chan School of Public Health Nutrition Department and this may be a valid asset either for determining patients with a very high oxalate intake and for reducing dietary oxalate intake in subjects at risk [84]. The non-linear association between dietary and urinary oxalate content was recently shown by Mitchell et al. There is a 2.7 mg increase in urinary oxalate excretion for every 100 mg of oxalate consumption, within a range of 50–750 mg/day and on a 1000 mg/day calcium diet [79]. However, even when dietary oxalate intake is lower than 50 mg/day, it significantly contributes to urinary oxalate excretion. This may be a consequence of the unsaturated intestinal oxalate transporters due to low intestinal oxalate concentration or because the majority of oxalate is soluble when oxalate intake in markedly reduced [1].

Calcium intake is equally relevant for determining urinary oxalate excretion; it chelates oxalate ions, thus reducing intestinal free oxalate ion availability and absorption [85,86]. This was demonstrated by a randomized trial of 120 patients with recurrent calcium oxalate kidney stones and hyperoxaluria comparing a low calcium diet (400 mg/day) to a low salt, low animal protein and balanced calcium diet (1200 mg/day) and showing a 60 μmol/day increase in urinary oxalate excretion in the low dietary calcium group [87].

#### 3.5.1. Nephrolithiasis and Nephrocalcinosis

Dietary modification is an effective first-line strategy in kidney stone prevention [88,89]. The effect of dietary oxalate intake on oxalate stone risk was investigated in a study conducted on three prospective, long-term cohorts of 45,985 men, 101,824 younger women, and 92,872 older women (Health Professionals Follow-up, Nurses’ Health Study I and II). Oxalate content of foods was assessed using the capillary electrophoresis method and dietary oxalate intake was assessed using food frequency questionnaires every four years, for a total combined follow-up of 44 years. The average oxalate consumption was 214 mg/day for men, 185 mg/day for older women, and 183 mg/day younger women respectively. Spinach accounted for more than 40% of total oxalate intake. Those in the highest quintile of dietary oxalate intake had only a modest increase of risk for incident kidney stones in multivariable models adjusted for age, body mass index, use of thiazide diuretics and other dietary factors [90]. Furthermore, no association was found between dietary oxalate intake and the risk for kidney stones in younger women. This may suggest that dietary oxalate alone is not a major risk factor for kidney stones. In contrast, cross-sectional studies in these same cohorts suggested a three to four-fold increased risk of prevalent kidney stones, when 24 h urine collections were taken into account [91]. In addition, a 5 mg increase in oxalate excretion was associated with a two-fold increase in the risk for kidney stones [91]. The different risk of stone formation calculated on estimated dietary oxalate intake and urinary excretion of oxalate might reflect the presence of other factors capable of influencing intestinal bioavailability and net absorption of oxalate. In fact, the proportion of dietary oxalate absorbed by the gut strongly depends on calcium intake: the association between oxalate consumption and its urinary excretion changes from approximately 20–40% to 50% if dietary calcium intake decreases from 1002 to 391 mg/day [1,92,93,94]. In addition, the exact estimate of dietary oxalate intake is affected by the intrinsic variability of oxalate content in foods. Different types of tea, for instance, contain varying levels of oxalate depending on specific characteristics such as harvesting period, brewing time, tea quality, and preparation [95,96,97].

The variability in intestinal oxalate absorption between stone formers versus non-stone formers should also be considered. Hesse et al. demonstrated that after an oral load of 50 mg of a (13C2)-labeled disodium oxalate salt, gut intestinal uptake of oxalate was significantly higher in patients with nephrolithiasis compared to healthy controls (9.2% versus 6.8%, respectively) [98]. In addition, higher animal protein intake seems to affect urinary oxalate excretion in idiopathic calcium stone formers but not in healthy subjects [99].

A better understanding of gut physiology revealed the importance of the gut microbiome in the intestinal handling of oxalate and the risk of calcium oxalate stone disease [11,100]. The intestinal colonization of *Oxalobacter formigenes,* an oxalate-degrading anaerobic bacterium, was associated with a lower urinary oxalate excretion [64]. Other commensals are also involved: one study conducted on 23 stone formers and 6 controls showed systematic differences in gut microbiome composition, in particular for *Bacteroides* (significantly more abundant in stone formers) and *Prevotella* (significantly more abundant in controls) [101].

Although increased urinary oxalate excretion and its dietary consumption are directly linked to kidney stones, oxalate crystal deposition in renal parenchyma is also responsible for nephrocalcinosis [102]. Mice fed with an oxalate-added spinach-derived extract developed crystalluria, hyperoxaluria and interstitial inflammation associated to nephrocalcinosis [103]. In humans affected by cystic fibrosis, the presence of absorptive hyperoxaluria has been linked to both nephrolithiasis and nephrocalcinosis [104,105].

#### 3.5.2. Acute Kidney Injury

Oxalate can cause acute kidney injury: acute oxalate nephropathy is defined as tubular obstruction and deposition by calcium-oxalate crystals in renal parenchyma [14,106]. The excess of crystals inside tubular cells may lead to different tissue injury: from phagolysosome overload and direct and indirect cytotoxic damage to oxalate-induced inflammation and necrosis [12,107]. In the majority of cases, it is caused by the ingestion of drugs or toxins metabolized to oxalate, such as ethylene glycol, vitamin C or orlistat [31,32,33]. In addition, it was hypothesized that different forms of oxalate deposition may influence disease manifestation: severe acute supersaturation may cause massive crystal deposition, renal epithelial cell damage, inflammation and necrosis, resulting in AKI. On the contrary, less severe but persistent forms may generate chronic crystal deposition, especially in the distal tubule or collecting ducts, ending in CKD [108,109]. Although prospective studies are lacking, a large number of case reports suggested a causative role of dietary oxalate on acute kidney injury. Star fruit-induced acute oxalate nephropathy is one of the most reported in literature [110,111,112,113]. A case series of 3 patients who presented with AKI after star fruit ingestion was recently published, suggesting the potential role of this fruit in inducing acute oxalate nephropathy in individuals with both normal and abnormal kidney function due to its high oxalate content [110], especially for fresh fruit juice (up to 829 mg/dL) [111]. A reported case of AKI in an 81 years old man with CKD and diabetes showed biopsy-proven oxalate nephropathy thought to be from the consumption of large amounts of vitamin C and high oxalate foods [52]. Other case-reports showed acute oxalate nephropathy linked to increased oxalate intake in the form of peanuts (> 130 g/day), green leaf-smoothies [114,115], rhubarb, and black iced-tea (16 8-oz glasses/day) [116]. It should be noted that the majority of the cases reported in the literature had other predisposing factors for AKI, such as dehydration, CKD, diabetes, and gastric bypass surgery (Table 2).

#### 3.5.3. Chronic Kidney Disease

Oxalate may also be a risk factor for CKD development or progression. In mice fed with a diet high in oxalate, a reproducible CKD phenotype developed with hypertension, hyperkalemia, metabolic acidosis, anemia, and hyperphosphatemia [119]. Renal histopathologic characteristics of these animals included interstitial fibrosis, tubular injury, atubular glomeruli, and inflammation. Consistently, tubular epithelial cell damage induced by tubular and interstitial calcium oxalate deposition, tubular atrophy and interstitial fibrosis have been described in humans [14]. In addition, oxalate alone has been recently associated with renal and systemic inflammation [15,120]. Its deposition in renal parenchyma resulted in activation of the innate immune system, driven by release of interleukin-1β via NLRP3 inflammasome (nucleotide-binding domain, leucine-rich repeat inflammasome 3) activation in dendritic cells and leading to progressive renal function decline in both acute and chronic animal models of tissue oxalate deposition [15,106,121]. Mulay et al. [121] also showed an indirect activation of NLRP3 inflammasome by oxalate-induced tubular cell damage and release of NLRP3 agonists. Furthermore, in a long-term mouse-model, mice fed with high oxalate diet developed both increased activation of NLRP3 inflammasome and interstitial inflammation in renal tissue surrounding calcium oxalate crystal deposition in renal tubules [106].

These data suggest the potential role of dietary oxalate in CKD progression in animal models [119].

Recently, the determinants of urinary oxalate and its association with more common forms of CKD were characterized in humans. In a large epidemiologic study of 3123 participants of the Chronic Renal Insufficiency Cohort (CRIC), Waikar et al. found that urinary oxalate correlated positively with proteinuria (r = 0.22, *p* < 0.001), and inversely with estimated glomerular filtration rate (eGFR) (r = −0.13, *p* < 0.001) and urinary calcium. Urinary oxalate excretion was associated with higher risk of CKD progression and ESRD, with the participants in the highest quintile of urinary oxalate having a 33% increased risk of CKD progression and 45% increased risk of ESRD compared to those in the lowest quintile (urinary oxalate ≥ 27.8 mg/day and <11.5 mg/day, respectively). The authors also found a non-linear relationship between urinary oxalate and renal outcomes, with those above the threshold effect (40th percentile) having a 32% increased risk of CKD progression and 37% increased risk of ESRD. Exploratory analyses suggested that diabetes and obesity modified the associations between urinary oxalate and renal outcomes, thus warranting further investigations [13]. The absolute values for oxalate excretion in this study should not be extrapolated to clinical use, because urine samples were stored for several years prior to measurement, with unknown effects on oxalate levels.

In primary hyperoxaluria patients with advanced kidney failure, plasma levels of oxalate may rise to extremely elevated concentrations (up to 125 μmol/L), reaching supersaturation limits [122,123]. On the contrary, in other forms of end stage renal disease, plasma oxalate concentration seems to be much lower [124,125]. In addition, given that oxalate endogenous production and extra-renal excretion are unchanged, reduced GFR does not affect the amount of oxalate excreted by the kidneys: at the steady state, an increase in plasma oxalate concentration is followed by higher filtered load and unchanged oxalate excretion rate [13]. Thus, the former described association between increased urinary oxalate excretion and lower eGFR may not be attributable to a CKD-related increase in plasma oxalate concentration.

Furthermore, while increasing evidence in recent years has implicated urinary oxalate in the development and progression of CKD, less is known about the role of dietary oxalate as a modifiable risk factor.

#### 3.5.4. Renal Transplantation

Whether oxalate excretion plays a factor in adverse outcomes following kidney transplantation is less clear [126]. Pinheiro et al. [127] studied kidney allograft biopsies from 97 transplant recipients and found that frequent calcium oxalate deposition was associated with more severe renal function decline and acute tubular necrosis. Graft survival after 12 years was significantly worse for patients with vs. without calcium oxalate deposition (49.7% vs. 74.1%, *p* = 0.013). In addition, tubule-interstitial fibrosis was significantly more frequent in patients with calcium oxalate deposition vs. controls [128].

The former associations were recently supported by Palsson et al. [129], who found a higher risk of delayed graft function in individuals with allograft biopsies showing calcium oxalate deposits. A few case series also reported early graft failure in patients with calcium oxalate deposition and increased urinary oxalate excretion [130].

In addition, there could be a different etiology in early versus late allograft dysfunction related to calcium oxalate deposition and increased urinary excretion of oxalate. During end stage renal disease, oxalate accumulates up to three times the normal range [122,131]. Immediately after kidney transplantation there is a remarkable excretion of retained calcium oxalate, progressively lowering plasma oxalate concentration [132] and perhaps creating a favorable condition for crystal deposition and tissue damage, especially if associated to other risk factors for early allograft dysfunction [133]. Calcium oxalate deposition may also cause both delayed graft function and faster long-term kidney function decline [129]. Furthermore, it is not clear whether the already reported evidence of reduced graft survival in case of calcium oxalate deposition yields different pathophysiology than the higher risk for CKD progression found in patients with higher urinary oxalate excretion [13]. The effect of immunosuppressive drugs as the mycophenolate-induced diarrhea may influence urinary oxalate excretion and its tissue deposition [134].

The association between oxalate excretion, outcomes in kidney transplantation and immunosuppressive treatment needs further study.

## 4. Conclusions

In this review, we suggest that excess oxalate has deleterious effects on kidney function. Oxalate is a terminal end product of metabolism in humans with no known role and its urinary supersaturation and crystal deposition in renal parenchyma have been associated with either stone formation, tubular cell damage, inflammation, tubular atrophy or interstitial fibrosis. This may play a role in the pathogenesis of conditions ranging from AKI to CKD to nephrocalcinosis, whereas in nephrolithiasis there is an established association between increased dietary intake and increased urinary excretion of oxalate with risk of kidney stones. Besides, we recognize that oxalate intake is not the only dietary factor involved in increased oxalate uptake: the amount of calcium intake strongly influences intestinal absorption of oxalate, suggesting a complex association between dietary oxalate consumption and its urinary excretion. Although dietary restriction of the highest oxalate foods and a balanced calcium diet remain a cornerstone of kidney stone formers management, the effect of reducing oxalate consumption on renal function decline in CKD patients and on the risk of acute kidney injury are not completely clear. In fact, hyperoxaluria can also clearly cause episodes of AKI. Ultimately, the association between oxalate excretion and increased risk of CKD progression deserves further study, as does oxalate in kidney transplantation. Additional mechanistic studies and prospective human studies will better define the relationship between oxalate and kidney diseases.

## Figures and Tables

**Table 1 nutrients-12-02673-t001:** Higher oxalate foods (mg of oxalate per 100 g serving).

Vegetables and Tubers	Cereals	Legumes	Nuts and Seeds	Others
Spinach (658)	Tofu (274)	Soybeans (77)	Almonds (407)	Amaranth (1090)
Rhubarb (433)	Rice bran (112)	Peas, raw (50)	Hazelnuts, raw (100)	Cocoa powder (111)
Beets (61)	Cornmeal (25.6)	Beans (25)	Peanuts (94.5)	Raspberries (19.2)
Potatoes (50)	Whole-grain flour (11.6)	Lentils, boiled (8)	Pistachio nuts (49)	Brewed tea (5.6)

Data from the Harvard T.H. Chan School of Public Health Nutrition Department and Han et al. [83].

**Table 2 nutrients-12-02673-t002:** Summary of case reports/case series of biopsy-proven oxalate nephropathy due to increased dietary oxalate consumption.

Year	Author	Involved Food	No. of Patients	Underlying CKD	Concomitant Risk Factors for AKI
2001	Chen et al. [111]	Star fruit	2	Normal	Dehydration (vomiting)
2012	Albersmeyer et al. [117]	Rhubarb	1	Normal	Dehydration (vomiting), type 1 diabetes
2014	Park et al. [114]	Peanut	1	Normal	Dehydration (vomiting and diarrhea)
2015	Syed et al. [116]	Black iced-tea	1	CKD G4	None
2017	Makkapati et al. [115]	Green leaf-smoothie	1	Normal	Type 2 diabetes, gastric bypass surgery
2018	Wijayaratne et al. [110]	Star fruit	3	Normal	Hypertension, type 2 diabetes
2019	Lin et al. [118]	Vitamin C supplements	1	Normal	Small bowel resection
2019	Clark et al. [52]	Vitamin C supplements, nuts, almonds, almond milk, cocoa powder, tomato paste, wheat germ	1	Normal	Type 2 diabetes

CKD: chronic kidney disease; AKI: acute kidney injury.

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
