# Peer review of "Dietary Oxalate Intake and Kidney Outcomes"

_nutrients, 2020, doi:10.3390/nu12092673_

Round 1

Reviewer 1 Report

The authors have provided a comprehensive review on the impact of oxalate on various kidney outcomes.  A review related to this topic has been published within the last year (Mitchell et. al. Am J Physiol Renal Physiol. 2019). However, the authors for this review have provided a more exhaustive review of the literature, focus on the role of oxalate on various kidney outcomes and not just stone disease, and highlight important areas that need further research.

Minor Comments

  1. Reference citations should be reevaluated in light of the first 2 paragraphs of Introduction citing inappropriate references- Line 29, reference 6, is not focused on hydroxyproline; Line 34, reference 10, is not an appropriate reference for gut microbiome dysbiosis.
  2. Section 3.2- although added in other parts of the review, reference to vitamin C as a source of endogenous oxalate synthesis should be added in this section
  3. The recent paper by Neumeier et al. JASN 2020 should be added to the review.

Author Response

REVIEWER 1

The authors have provided a comprehensive review on the impact of oxalate on various kidney outcomes.  A review related to this topic has been published within the last year (Mitchell et. al. Am J Physiol Renal Physiol. 2019). However, the authors for this review have provided a more exhaustive review of the literature, focus on the role of oxalate on various kidney outcomes and not just stone disease, and highlight important areas that need further research.

Minor Comments

  1. Reference citations should be reevaluated in light of the first 2 paragraphs of Introduction citing inappropriate references- Line 29, reference 6, is not focused on hydroxyproline; Line 34, reference 10, is not an appropriate reference for gut microbiome dysbiosis.

REPLY: We removed reference 10 and updated reference focused on hydroxyproline (L29)

2. Section 3.2- although added in other parts of the review, reference to vitamin C as a source of endogenous oxalate synthesis should be added in this section

REPLY: We added the suggested information in section 3.2, as follow: L85-86 In addition, ingestion of a variety of substances, including ethylene glycol, vitamin C or orlistat are metabolized to oxalate

3. The recent paper by Neumeier et al. JASN 2020 should be added to the review.

REPLY: We added the suggested paper to the manuscript: L71-72: Furthemore, an increased intestinal oxalate excretion mediated by higher SLC26A6 activity was recently found in a mouse model of CKD

Reviewer 2 Report

The review entitled, “Dietary oxalate intake and kidney outcomes” is a well written and organized manuscript summarizing a large portion of the current literature connecting dietary oxalate intake to major kidney outcomes, specifically nephrolithiasis, AKI, and CKD. The authors provide a well-rounded summary of oxalate-induced renal dysfunction at several different stages; however, major and minor limitations exist which should be addressed:

Major:

  1. Information in Table 1 needs to be further explained in the text, i.e. from where was this data gathered; how was it analyzed; how does it factor into the conclusions presented in Section 3.5? Additionally, conclusions drawn in this section are contradictory to what one would expect in terms of urinary oxalate excretion with increased levels of dietary oxalate intake. The authors should consider expanding this section to further explain paradoxical effects of lower dietary intake correlating to higher urinary oxalate excretion, as well as the role of calcium homeostasis when oxalate concentrations are in flux.
  2. Sections 3.5.2 and 3.5.3 discuss the role of dietary oxalate-induced AKI and CKD, respectively, but lack sufficient distinction and follow-up explanation(s) between the toxicity of calcium oxalate vs. the oxalate ion itself. Oxalate ion-induced renal dysfunction differs slightly mechanistically than that of calcium oxalate crystal-induced mechanical damage and subsequent AKI and/or CKD, and these details should be further delineated in both sections.
  3. Table 2 needs an organizational component associated to assist with reader-acquisition: whether these case reports be listed chronologically, by number of most to least patients affected, or alphabetically.
  4. The Conclusions section is a recapitulation of the abstract, and lacks appropriate, succinct conclusions based upon what was discussed in the body of the review. A major revision of this section should be considered to include specific foci of the perceived association between dietary oxalate and secondary kidney outcomes discussed in the review.

Minor:

  1. The nomenclature of the SLC transporters is inconsistent throughout the manuscript, e.g. upper case in some areas, lower case in others, etc. When discussing genes which encode specific solute carrier transporter proteins, these should be italicized; whereas, when discussing the resultant protein, it should remain initialized.
  2. Line 74 should read, “reduction in urinary oxalate excretion”
  3. Line 122 should read, “each caused by a gene mutation in an enzyme involved in”
  4. When discussing the role of O. formigenes in Section 3.4, genus and species should be italicized throughout.
  5. The last sentence of Section 3.5 seems misplaced and should be deleted.
  6. When discussing tubular epithelial cell damage in section 3.5.3, cells should not be pluralized, refer to lines 247 and 254, specifically.

Author Response

REVIEWER 2

The review entitled, “Dietary oxalate intake and kidney outcomes” is a well written and organized manuscript summarizing a large portion of the current literature connecting dietary oxalate intake to major kidney outcomes, specifically nephrolithiasis, AKI, and CKD. The authors provide a well-rounded summary of oxalate-induced renal dysfunction at several different stages; however, major and minor limitations exist which should be addressed:

Major:

  1. Information in Table 1 needs to be further explained in the text, i.e. from where was this data gathered; how was it analyzed; how does it factor into the conclusions presented in Section 3.5? Additionally, conclusions drawn in this section are contradictory to what one would expect in terms of urinary oxalate excretion with increased levels of dietary oxalate intake. The authors should consider expanding this section to further explain paradoxical effects of lower dietary intake correlating to higher urinary oxalate excretion, as well as the role of calcium homeostasis when oxalate concentrations are in flux

REPLY: As suggested, we added information about table 1 and further explained the evidence shown in section 3.5 (L165-170: In detail, the exact oxalate content of foods is equally determinable with both capillary electrophoresis and ion chromatography. Examples of high oxalate foods are green-leaf vegetables, tea, nuts, chocolate and rhubarb. The list of higher oxalate foods reported in table 1 was obtained using data shown by the Harvard T.H. Chan School of Public Health Nutrition Department's and it may be a valid asset either for determining patients with a very high oxalate intake and for reducing dietary oxalate intake in subjects at risk. L173-180: However, even when dietary oxalate intake is lower than 50 mg/day, it significantly contributes to urinary oxalate excretion. This may be a consequence of the unsaturated intestinal oxalate transporters due to low intestinal oxalate concentration or because the majority of oxalate is soluble when oxalate intake in markedly reduced. Calcium intake is equally relevant for determining urinary oxalate excretion; it chelates oxalate ions, thus reducing intestinal free oxalate ions availability and absorption.)

.

2. Sections 3.5.2 and 3.5.3 discuss the role of dietary oxalate-induced AKI and CKD, respectively, but lack sufficient distinction and follow-up explanation(s) between the toxicity of calcium oxalate vs. the oxalate ion itself. Oxalate ion-induced renal dysfunction differs slightly mechanistically than that of calcium oxalate crystal-induced mechanical damage and subsequent AKI and/or CKD, and these details should be further delineated in both sections.

REPLY: we further added pathophysiological distinction between oxalate-induced AKI and CKD. As regards calcium-oxalate and oxalate ion renal injury, we already explained in CKD section the effect of oxalate on renal inflammation. Anyway, we better clarified the different causes of renal injury involved, in both AKI and CKD sections (L231-240: Oxalate can cause acute kidney injury: acute oxalate nephropathy is defined as tubular obstruction and deposition by calcium-oxalate crystals in renal parenchyma. The excess of crystals inside tubular cells may lead to different tissue injury: from phagolysosome overload, direct and indirect cytotoxic damage to oxalate-induced inflammation and necrosis……… In addition, it was hypothesized that different forms of oxalate deposition may influence disease manifestation: severe acute supersaturation may cause massive crystals deposition, renal epithelial cell damage, inflammation and necrosis, resulting in AKI. On the contrary, less severe but persistent forms may generate chronic crystals deposition, especially in the distal tubule or collecting ducts, ending in CKD. L262: In addition, oxalate alone has been recently associated to renal and systemic inflammation)

3. Table 2 needs an organizational component associated to assist with reader-acquisition: whether these case reports be listed chronologically, by number of most to least patients affected, or alphabetically.

REPLY: we listed table 2 in chronological order.

4. The Conclusions section is a recapitulation of the abstract, and lacks appropriate, succinct conclusions based upon what was discussed in the body of the review. A major revision of this section should be considered to include specific foci of the perceived association between dietary oxalate and secondary kidney outcomes discussed in the review.

REPLY: we changed the conclusion section as follow: L326-341 (In this review, we suggest that excess oxalate has deleterious effects on kidney function. Oxalate is a terminal end product of metabolism in humans with no known role and its urinary supersaturation and crystal deposition in renal parenchyma have been associated to either stone formation, tubular cell damage, inflammation, tubular atrophy and interstitial fibrosis. This may play a role in the pathogenesis of conditions ranging from AKI to CKD to nephrocalcinosis, whereas in nephrolithiasis there is an established association between increased dietary intake and increased urinary excretion of oxalate with risk of kidney stones. Besides, we recognize that oxalate intake is not the only dietary factor involved in increased oxalate uptake: the amount of calcium intake strongly influences intestinal absorption of oxalate, suggesting a complex association between dietary oxalate consumption and its urinary excretion. Although dietary restriction of the highest oxalate foods and a balanced calcium diet remain a cornerstone of kidney stones formers management, the effect of reducing oxalate consumption on renal function decline in CKD patients and on the risk of acute kidney injury are not completely clear. In fact, hyperoxaluria can also clearly cause episodes of AKI. Ultimately, the association between oxalate excretion and increased risk of CKD progression deserves further study, as does oxalate in kidney transplantation. Additional mechanistic studies and prospective human studies will better define the relationship between oxalate and kidney diseases.)

Minor:

  1. The nomenclature of the SLC transporters is inconsistent throughout the manuscript, e.g. upper case in some areas, lower case in others, etc. When discussing genes which encode specific solute carrier transporter proteins, these should be italicized; whereas, when discussing the resultant protein, it should remain initialized.

REPLY: We changed the terminology of SLC transporters in every section of the manuscript, as suggested.

2. Line 74 should read, “reduction in urinary oxalate excretion”

REPLY: sentence changed as suggested

3. Line 122 should read, “each caused by a gene mutation in an enzyme involved in”

REPLY: sentence changed as suggested

4. When discussing the role of O. formigenes in Section 3.4, genus and species should be italicized throughout.

REPLY: we italicized genus and species of O. formigenes in all the manuscript as suggested

5. The last sentence of Section 3.5 seems misplaced and should be deleted.

REPLY: sentence removed

6. When discussing tubular epithelial cell damage in section 3.5.3, cells should not be pluralized, refer to lines 247 and 254, specifically.

REPLY: sentence changed as suggested

Reviewer 3 Report

Overall good review of the role of oxalate in renal disease. The authors should go into more detail describing the bonding of calcium and oxalate in the gastrointestinal tract to explain why increased dietary calcium reduces oxalate excretion in the urine. There should also be some mention of new medications on the horizon to treat primary hyperoxaluria. 

Author Response

REVIEWER 3

Overall good review of the role of oxalate in renal disease. The authors should go into more detail describing the bonding of calcium and oxalate in the gastrointestinal tract to explain why increased dietary calcium reduces oxalate excretion in the urine. There should also be some mention of new medications on the horizon to treat primary hyperoxaluria. 

REPLY: Thank you for these suggestions. We have added these sections accordingly under section 3.4. (L147-152: Dietary calcium has been shown to affect urinary oxalate excretion. Early in vitro experiments have demonstrated that calcium binds to oxalate causing it to be insoluble. In vivo, increased calcium supplementation significantly reduced urinary oxalate excretion, leading to the postulate of calcium precipitating oxalate and thus preventing its absorption in the gut. Several epidemiologic studies have also shown the inverse relationship between dietary calcium intake and urinary oxalate excretion in both healthy individuals and stone formers. L164-177: While orthotopic liver transplantation is the only definitive cure for PH, numerous potential therapies for primary and enteric hyperoxalurias are currently being studied. One approach to PH therapy involves downregulation of key enzymes—glycolate oxidase and lactate dehydrogenase, in the oxalate metabolism pathway through gene silencing methods or enzyme inhibition. Stiripentol is a non-competitive inhibitor of lactate dehydrogenase 5 enzyme that has been shown recently to reduce hepatic production of oxalate and subsequent urinary oxalate excretion in rats and in humans. A phase 2 clinical trial is currently underway in evaluating the efficacy of Stiripentol in patients with PH (https://clinicaltrials.gov/ct2/show/NCT03819647). Another approach involves oral administration of O. formigenes to increase gut degradation of oxalate thereby decreasing subsequent oxalate absorption (https://clinicaltrials.gov/ct2/show/NCT03116685). For enteric hyperoxaluria, oral administration of oxalate decarboxylase is currently being studied by two pharmaceutical companies with promising results (https://clinicaltrials.gov/ct2/show/NCT03847090). An in-depth review of novel investigational therapies for the hyperoxalurias was recently published by Kletzmayr and colleagues)